# Smooth operator: Modifying the Anhøj rules to improve runs analysis in statistical process control

Jacob Anhøj[1]*, Tore Wentzel-Larsen[2]

**1** Centre of Diagnostic Investigation, Rigshospitalet, University of Copenhagen, Copenhagen, Denmark,
**2** Centre for Child and Adolescent Mental Health, Eastern and Southern Norway & Centre for Violence and Traumatic Stress Studies, Oslo, Norway

☯ These authors contributed equally to this work.
* jacob@anhoej.net

**Data Availability Statement:** All relevant data are within the paper and its Supporting Information files.

**Funding:** The authors received no specific funding for this work.

## Abstract

### Introduction

The run chart is one form of statistical process control chart that is particularly useful for detecting persistent shifts in data over time. The Anhøj rules test for shifts by looking for unusually long runs (L) of data points on the same side of the process centre (mean or median) and unusually few crossings (C) of the centre depending on the number of available data points (N). Critical values for C and L have mainly been studied in isolation. But what is really of interest is the joint distribution of C and L, which has so far only been studied using simulated data series. We recently released an R package, crossrun that calculates exact values for the joint probabilities of C and L that allowed us to study the diagnostic properties of the Anhøj rules in detail and to suggest minor adjustments to improve their diagnostic value.

### Methods

Based on the crossrun R package we calculated exact values for the joint distribution of C and L for N = 10–100. Furthermore, we developed two functions, bestbox() and cutbox() that automatically seek to adjust the critical values for C and L to balance between sensitivity and specificity requirements.

### Results

Based on exact values for the joint distribution of C and L for N = 10–100 we present measures of the diagnostic value of the Anhøj rules. The best box and cut box procedures improved the diagnostic value of the Anhøj rules by keeping the specificity and sensitivity close to pre-specified target values.

**Competing interests:** The authors have declared that no competing interests exist.

## Conclusions

Based on exact values for the joint distribution of longest run and number of crossings in random data series this study demonstrates that it is possible to obtain better diagnostic properties of run charts by making minor adjustment to the critical values for C and L.

## Introduction

Within statistical process control (SPC) runs analysis is being used to detect persistent shifts in process location over time [1].

Runs analysis deals with the natural limits of number of runs and run lengths in random processes. A run is a series of one or more consecutive elements of the same kind, for example heads and tails, diseased and non-diseased individuals, or numbers above or below a certain value. A run chart is a point-and-line chart showing data over time with the median as reference line (Fig 1). In a random process, the data points will be randomly distributed around the median, and the number and lengths of runs will be predictable within limits. All things being equal, if the process shifts, runs tend to become longer and fewer. Consequently, runs analysis may help detect shifts in process location. Process shifts are a kind of non-random variation in time series data that are of particular interest to quality control and improvement: If a process shifts, it may be the result of planned improvement or unwanted deterioration.

Several tests (or rules) based on the principles of runs analysis for detection of shifts exist. In previous papers we demonstrated, using simulated data series, that the currently best performing rules with respect to sensitivity and specificity to shifts in process location are two simple tests [1–3]:

- Shifts test: one or more unusually long runs of data points on the same side of the centre line.

- Crossings test: the curve crosses the centre line unusually few times.

Collectively, we refer to these tests as the Anhøj rules, which are the default rules used for run and control chart analysis with the qicharts2 package for R [4]. For a thorough discussion of the practical use of run and control charts for quality improvement we refer to the qicharts2 package vignette.

Critical values for run length and number of crossings depend on the total number of data points in the chart, excluding data points that fall directly on the centre line. The number of crossings follows a binomial distribution, $b(N- 1, 0.5)$, where N is the number of data points and 0.5 the success probability. Thus, the lower prediction limit for number of crossings may, for example, be set to the lower 5th percentile of the corresponding cumulative binomial distribution [5]. However, no closed form expression exists for the distribution of longest runs. Consequently, the upper prediction limit for longest runs has traditionally been either a fixed value (usually 7 or 8) [6] or an approximate value depending on N as with the Anhøj rules: $\log_2(N) + 3$ rounded to the nearest integer [7]. Fig 1 has 20 data points, the curve crosses the centre line 9 times, and the longest run (points 3–6) contains 4 data points. In a random process with 20 data points, we should expect at least 6 crossings and the longest run should include no more than 7 data points. Thus, according to the Anhøj rules, Fig 1 shows random variation.

Each of the two tests has an overall specificity (true negative proportion) around 95%. The sensitivity (true positive proportion) of a test depends on the size of the shift (signal) relative to

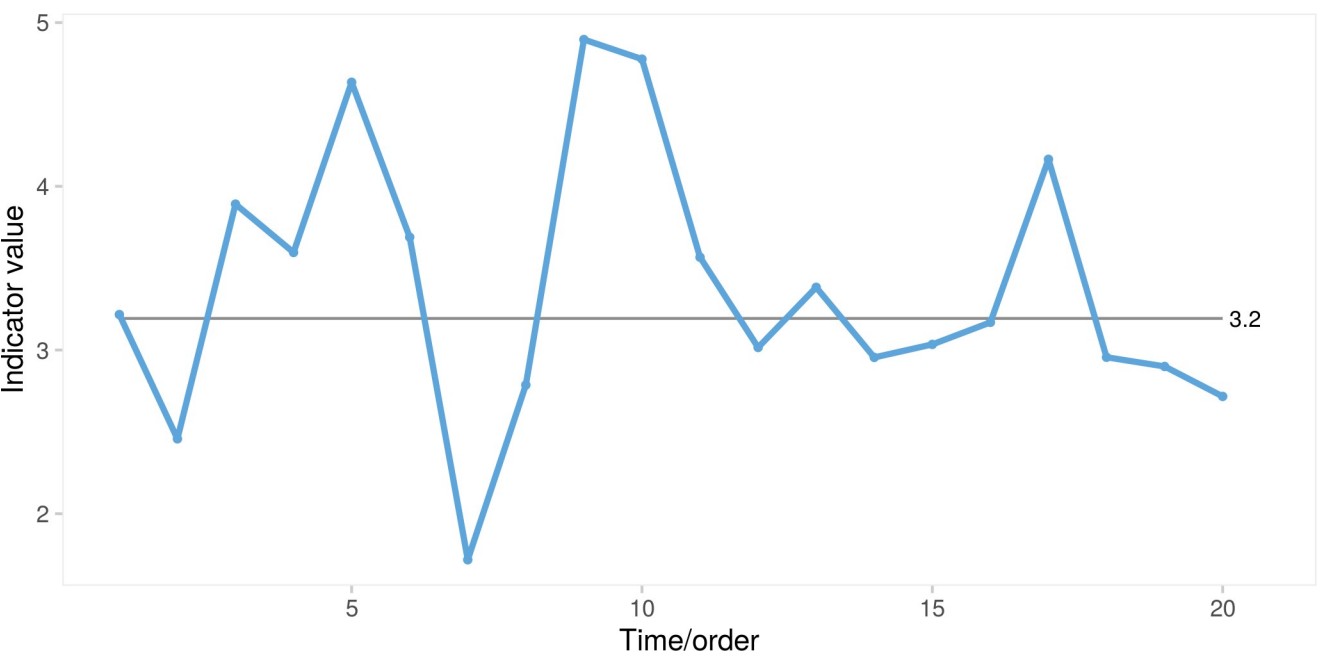

**Fig 1. Run chart.** Median = 3.2, longest run (L) = 4, number of crossings (C) = 9.

the random variation inherent in the process (noise). When applied together, the sensitivity increases, while the specificity decreases a bit and fluctuates around 92.5% (see red line in Fig 2).

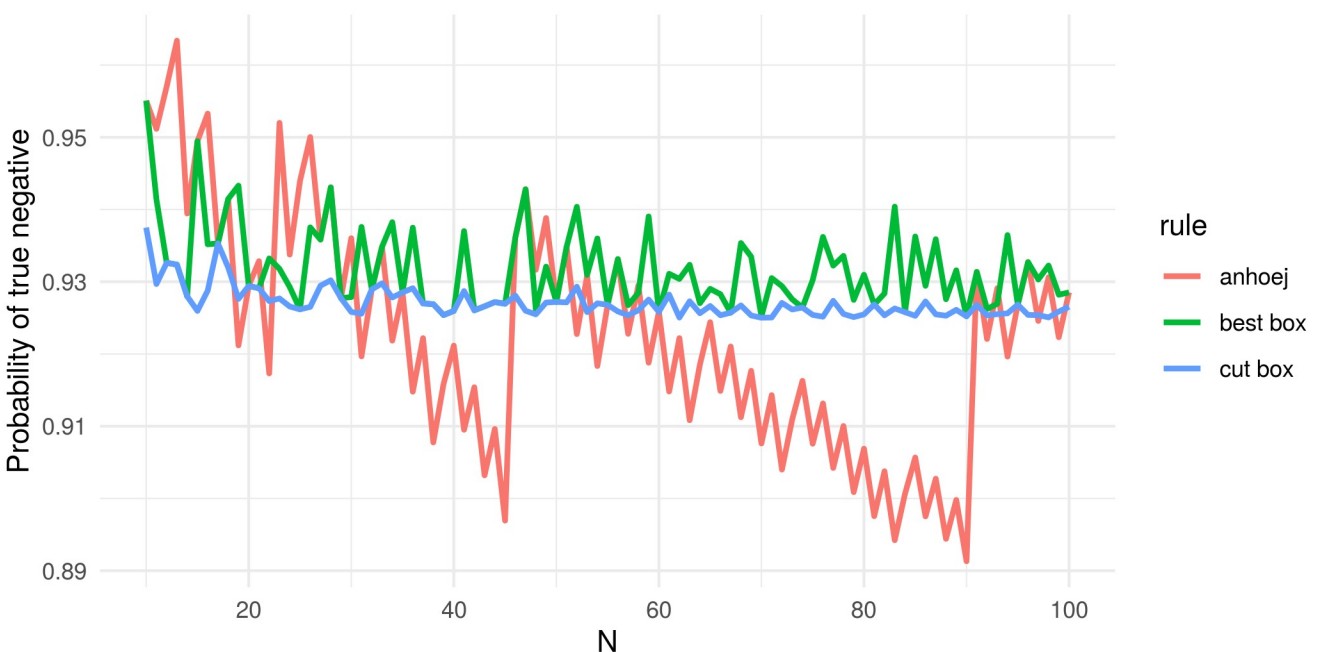

**Fig 2. Specificity of the Anhøj, best box, and cut box rules.** N = number of data points in run chart.

Historically, runs tests have been studied in isolation. But what is really of interest because the rules are linked–when runs grow longer, crossings become fewer–is the properties of the joint distribution of number of crossings (C) and longest runs (L).

We recently released an R package, crossrun [8,9], that includes functions for calculating the joint probabilities of C and L in random data series of different lengths (N) and with and without shifts in process location expressed in standard deviation units (SD). Fig 3 illustrates this for a run chart with N = 11 and SD = 0 (no shift). To avoid very small numbers, the probabilities are shown using the times representation, that is, the probabilities times $2^{N-1}$, which is 1024 for N = 11. The red box encloses the combinations of C and L that would indicate random variation according to the Anhøj rules (true negatives). The area outside the box represents combinations of C and L that would indicate non-random variation (false positives).

With the crossrun package it became feasible to calculate exact joint probabilities of C and L over a range of N and SD. And consequently, it became feasible to investigate the diagnostic properties of run charts using exact values for specificity and sensitivity rather than values based on time consuming, inaccurate, and complicated simulation studies.

As shown in Fig 2 the specificity of the Anhøj rules (red line) jumps up and down as N changes. This is a consequence of the discrete nature of the two tests–especially the shifts test. Although the specificity of the Anhøj rules does not decrease continuously as N increases, which is the case for other rules [2], we hypothesised that it would be possible to improve the diagnostic value further by smoothing the specificity using minor adjustments to C and L depending on N.

The aims of this study were to provide exact values for the diagnostic properties of the Anhøj rules and to suggest a "smoothing" procedure for improving the value of runs analysis.

## Methods

### Likelihood ratios to quantify the diagnostic value of runs rules

The value of diagnostic tests has traditionally been described using terms like sensitivity and specificity. These parameters express the probability of detecting the condition being tested for when it is present and not detecting it when it is absent:

Specificity = P(no signal | no shift) = P(true negative) = 1 –P(false positive)

Sensitivity = P(signal | shift) = P(true positive) = 1 –P(false negative)

For example, the specificity of the Anhøj rules in a run chart with 11 data points may be calculated from Fig 3 as the proportion enclosed by the red box, which is 974 / 1024 = 0.9512. The sensitivity may be obtained from a similar matrix (not shown) including a shift as the proportion being outside the box. With a shift of 0.8 SD, the sensitivity is 0.3493 (Table 1).

However, we usually seek to answer the opposite question: what is the likelihood that a positive or negative test actually represents the condition being tested for? Likelihood ratios (LR) do this:

LR+ = TP / FP = sensitivity / (1 –specificity)

LR– = FN / TN = (1 –sensitivity) / specificity

Accordingly, with 11 data points and a shift of 0.8 SD, LR+ = 0.3493 / (1–0.9512) = 7.2, and LR- = (1–0.3493) / 0.9512 = 0.68.

Detailed explanations of likelihood ratios have been given previously [3,10]. As stated in [3], a likelihood ratio greater than 1 speaks in favour of the condition being tested for, and a likelihood ratio less than 1 speaks against the condition. As a rule of thumb, also presented in [3], a positive likelihood ratio (LR+) greater than 10 is described as strong evidence that the condition is present, and a negative likelihood ratio (LR–) smaller than 0.1 is described as

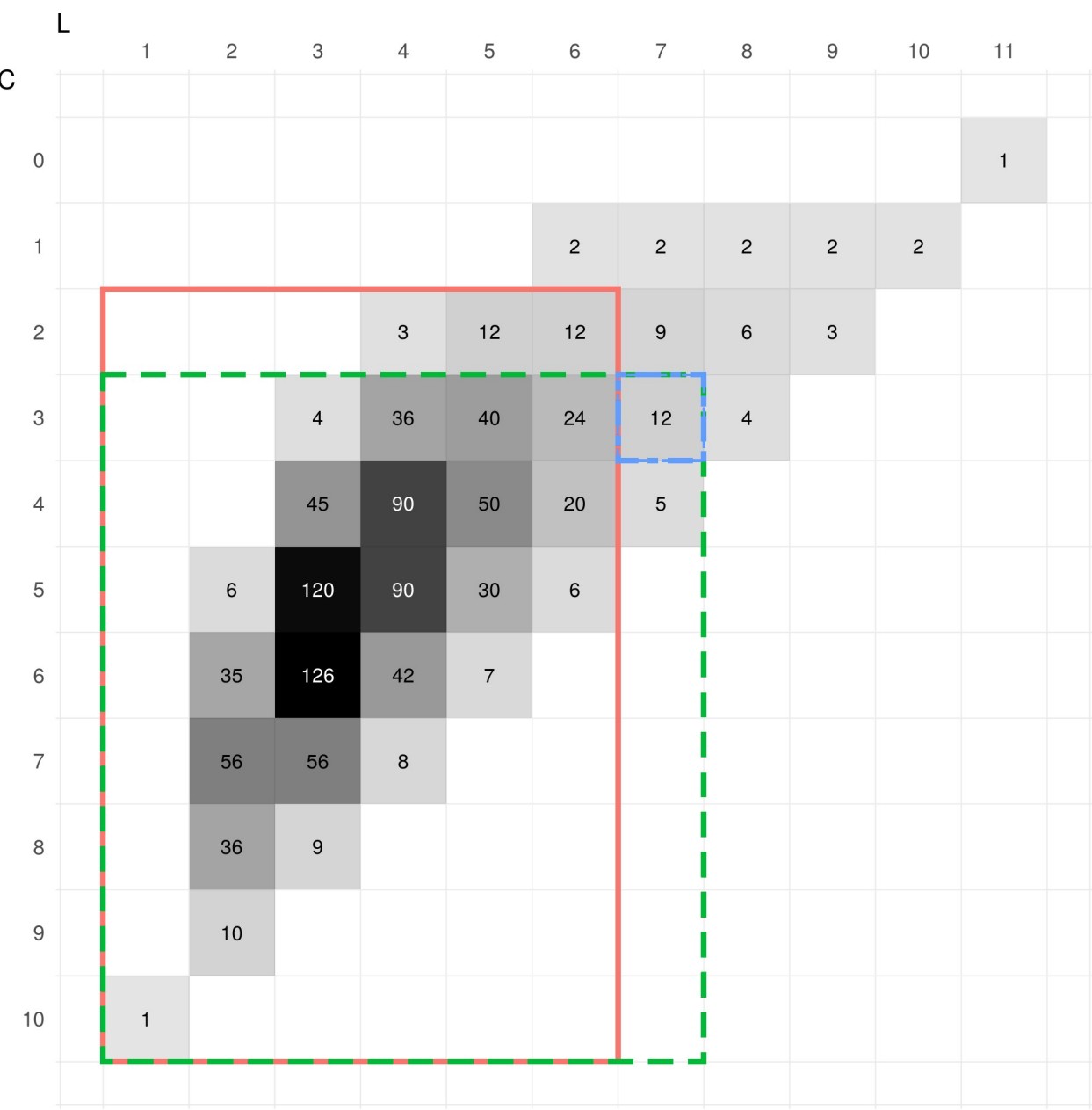

**Fig 3. Borders of the Anhøj, best box, and cut box rules for N = 11 data points.**

strong evidence against the condition [10]. For example, if LR+ = 5 and LR− = 0.2, a positive test means that it is 5 times *more* likely that the condition is present than not present, and a negative test means that it is 5 times *less* likely that the condition is present than not present. Thus, as detailed in [3], likelihood ratios always occur in pairs and together constitute combined measures of the usefulness of a diagnostic test. Specifically, for our purpose, run charts are diagnostic tests for non-random variation in time series data [1,3].

**Table 1. Signal limits and diagnostic values of the Anhøj, best box, and cut box rules.**

| | Anhøj | | Best box | | Cut box | | Specificity | | | Sensitivity | | |
|---|---|---|---|---|---|---|---|---|---|---|---|---|
| N | C | L | C | L | Cbord | Lbord | Anhøj | Best box | Cut box | Anhøj | Best box | Cut box |
| 10 | 2 | 6 | 2 | 6 | 3 | 5 | 0.9551 | 0.9551 | 0.9375 | 0.3103 | 0.3103 | 0.3786 |
| 11 | 2 | 6 | 3 | 7 | 4 | 6 | 0.9512 | 0.9414 | 0.9297 | 0.3493 | 0.3887 | 0.4211 |
| 12 | 3 | 7 | 3 | 6 | | | 0.9570 | 0.9326 | 0.9326 | 0.3677 | 0.4392 | 0.4392 |
| 13 | 3 | 7 | 3 | 6 | | | 0.9634 | 0.9324 | 0.9324 | 0.3628 | 0.4519 | 0.4519 |
| 14 | 4 | 7 | 3 | 6 | | | 0.9395 | 0.9280 | 0.9280 | 0.4051 | 0.4740 | 0.4740 |
| 15 | 4 | 7 | 4 | 7 | 6 | 6 | 0.9495 | 0.9495 | 0.9260 | 0.4046 | 0.4046 | 0.4806 |
| 16 | 4 | 7 | 5 | 8 | 6 | 7 | 0.9533 | 0.9352 | 0.9288 | 0.4146 | 0.4800 | 0.4993 |
| 17 | 5 | 7 | 5 | 7 | | | 0.9353 | 0.9353 | 0.9353 | 0.5069 | 0.5069 | 0.5069 |
| 18 | 5 | 7 | 5 | 7 | 6 | 6 | 0.9415 | 0.9415 | 0.9320 | 0.5030 | 0.5030 | 0.5256 |
| 19 | 6 | 7 | 5 | 7 | 6 | 5 | 0.9212 | 0.9433 | 0.9276 | 0.5370 | 0.5078 | 0.5351 |
| 20 | 6 | 7 | 6 | 7 | | | 0.9294 | 0.9294 | 0.9294 | 0.5372 | 0.5372 | 0.5372 |
| 21 | 6 | 7 | 7 | 8 | | | 0.9328 | 0.9291 | 0.9291 | 0.5447 | 0.5672 | 0.5672 |
| 22 | 7 | 7 | 6 | 7 | 7 | 6 | 0.9173 | 0.9332 | 0.9273 | 0.6121 | 0.5573 | 0.5902 |
| 23 | 7 | 8 | 6 | 7 | 7 | 6 | 0.9520 | 0.9318 | 0.9277 | 0.5322 | 0.5728 | 0.5983 |
| 24 | 8 | 8 | 6 | 7 | 7 | 6 | 0.9338 | 0.9293 | 0.9266 | 0.5646 | 0.5900 | 0.6084 |
| 25 | 8 | 8 | 6 | 7 | | | 0.9439 | 0.9262 | 0.9262 | 0.5536 | 0.6077 | 0.6077 |
| 26 | 8 | 8 | 9 | 9 | 10 | 7 | 0.9500 | 0.9375 | 0.9265 | 0.5488 | 0.5986 | 0.6298 |
| 27 | 9 | 8 | 9 | 8 | 10 | 7 | 0.9358 | 0.9358 | 0.9295 | 0.6221 | 0.6221 | 0.6397 |
| 28 | 9 | 8 | 9 | 8 | 11 | 7 | 0.9431 | 0.9431 | 0.9302 | 0.6118 | 0.6118 | 0.6589 |
| 29 | 10 | 8 | 10 | 8 | | | 0.9277 | 0.9277 | 0.9277 | 0.6382 | 0.6382 | 0.6382 |
| 30 | 10 | 8 | 11 | 10 | 12 | 9 | 0.9360 | 0.9279 | 0.9258 | 0.6299 | 0.6533 | 0.6617 |
| 31 | 11 | 8 | 11 | 9 | 14 | 8 | 0.9197 | 0.9376 | 0.9256 | 0.6958 | 0.6515 | 0.6880 |
| 32 | 11 | 8 | 11 | 8 | | | 0.9289 | 0.9289 | 0.9289 | 0.6843 | 0.6843 | 0.6843 |
| 33 | 11 | 8 | 11 | 8 | 12 | 7 | 0.9348 | 0.9348 | 0.9298 | 0.6766 | 0.6766 | 0.6912 |
| 34 | 12 | 8 | 11 | 8 | 13 | 7 | 0.9218 | 0.9382 | 0.9278 | 0.6982 | 0.6724 | 0.7141 |
| 35 | 12 | 8 | 12 | 8 | | | 0.9285 | 0.9285 | 0.9285 | 0.6920 | 0.6920 | 0.6920 |
| 36 | 13 | 8 | 13 | 9 | 15 | 8 | 0.9148 | 0.9375 | 0.9291 | 0.7442 | 0.6966 | 0.7265 |
| 37 | 13 | 8 | 14 | 10 | | | 0.9222 | 0.9270 | 0.9270 | 0.7356 | 0.6940 | 0.6940 |
| 38 | 14 | 8 | 13 | 8 | | | 0.9078 | 0.9269 | 0.9269 | 0.7548 | 0.7298 | 0.7298 |
| 39 | 14 | 8 | 15 | 11 | | | 0.9158 | 0.9254 | 0.9254 | 0.7475 | 0.7308 | 0.7308 |
| 40 | 14 | 8 | 15 | 9 | | | 0.9212 | 0.9260 | 0.9260 | 0.7430 | 0.7509 | 0.7509 |
| 41 | 15 | 8 | 15 | 9 | 17 | 8 | 0.9095 | 0.9370 | 0.9287 | 0.7846 | 0.7353 | 0.7642 |
| 42 | 15 | 8 | 14 | 8 | | | 0.9154 | 0.9260 | 0.9260 | 0.7782 | 0.7408 | 0.7408 |
| 43 | 16 | 8 | 14 | 8 | | | 0.9032 | 0.9266 | 0.9266 | 0.7938 | 0.7427 | 0.7427 |
| 44 | 16 | 8 | 17 | 10 | | | 0.9096 | 0.9272 | 0.9272 | 0.7884 | 0.7704 | 0.7704 |
| 45 | 17 | 8 | 17 | 9 | | | 0.8969 | 0.9270 | 0.9270 | 0.8249 | 0.7815 | 0.7815 |
| 46 | 17 | 9 | 17 | 9 | 19 | 8 | 0.9361 | 0.9361 | 0.9281 | 0.7687 | 0.7687 | 0.7961 |
| 47 | 17 | 9 | 17 | 9 | 20 | 7 | 0.9428 | 0.9428 | 0.9260 | 0.7576 | 0.7576 | 0.8045 |
| 48 | 18 | 9 | 19 | 12 | 20 | 11 | 0.9317 | 0.9261 | 0.9255 | 0.7750 | 0.7863 | 0.7896 |
| 49 | 18 | 9 | 19 | 10 | 21 | 9 | 0.9388 | 0.9321 | 0.9271 | 0.7648 | 0.7928 | 0.8099 |
| 50 | 19 | 9 | 19 | 9 | | | 0.9272 | 0.9272 | 0.9272 | 0.8082 | 0.8082 | 0.8082 |
| 51 | 19 | 9 | 19 | 9 | 21 | 8 | 0.9348 | 0.9348 | 0.9271 | 0.7976 | 0.7976 | 0.8233 |
| 52 | 20 | 9 | 19 | 9 | 21 | 7 | 0.9228 | 0.9404 | 0.9293 | 0.8131 | 0.7885 | 0.8238 |
| 53 | 20 | 9 | 21 | 11 | 23 | 9 | 0.9308 | 0.9310 | 0.9258 | 0.8034 | 0.8120 | 0.8292 |
| 54 | 21 | 9 | 21 | 10 | 23 | 8 | 0.9183 | 0.9360 | 0.9270 | 0.8413 | 0.8130 | 0.8385 |
| 55 | 21 | 9 | 21 | 9 | | | 0.9268 | 0.9268 | 0.9268 | 0.8315 | 0.8315 | 0.8315 |

(*Continued*)

**Table 1.** (Continued)

| | Anhøj | | Best box | | Cut box | | Specificity | | | Sensitivity | | |
|---|---|---|---|---|---|---|---|---|---|---|---|---|
| N | C | L | C | L | Cbord | Lbord | Anhøj | Best box | Cut box | Anhøj | Best box | Cut box |
| 56 | 21 | 9 | 21 | 9 | 23 | 8 | 0.9331 | 0.9331 | 0.9259 | 0.8228 | 0.8228 | 0.8465 |
| 57 | 22 | 9 | 23 | 12 | 25 | 11 | 0.9228 | 0.9268 | 0.9254 | 0.8360 | 0.8341 | 0.8403 |
| 58 | 22 | 9 | 23 | 10 | 24 | 9 | 0.9295 | 0.9285 | 0.9260 | 0.8280 | 0.8441 | 0.8506 |
| 59 | 23 | 9 | 23 | 10 | 26 | 8 | 0.9188 | 0.9390 | 0.9275 | 0.8600 | 0.8312 | 0.8595 |
| 60 | 23 | 9 | 23 | 9 | | | 0.9258 | 0.9258 | 0.9258 | 0.8520 | 0.8520 | 0.8520 |
| 61 | 24 | 9 | 23 | 9 | 24 | 8 | 0.9148 | 0.9311 | 0.9282 | 0.8636 | 0.8448 | 0.8529 |
| 62 | 24 | 9 | 25 | 11 | 27 | 9 | 0.9222 | 0.9304 | 0.9250 | 0.8560 | 0.8552 | 0.8703 |
| 63 | 25 | 9 | 25 | 10 | 27 | 9 | 0.9108 | 0.9323 | 0.9273 | 0.8839 | 0.8588 | 0.8732 |
| 64 | 25 | 9 | 26 | 11 | 27 | 10 | 0.9185 | 0.9270 | 0.9256 | 0.8766 | 0.8558 | 0.8628 |
| 65 | 25 | 9 | 26 | 10 | 27 | 9 | 0.9244 | 0.9290 | 0.9266 | 0.8699 | 0.8606 | 0.8709 |
| 66 | 26 | 9 | 27 | 12 | 29 | 10 | 0.9149 | 0.9283 | 0.9254 | 0.8798 | 0.8701 | 0.8796 |
| 67 | 26 | 9 | 27 | 10 | | | 0.9210 | 0.9257 | 0.9257 | 0.8736 | 0.8820 | 0.8820 |
| 68 | 27 | 9 | 27 | 10 | 29 | 8 | 0.9112 | 0.9354 | 0.9267 | 0.8973 | 0.8720 | 0.8923 |
| 69 | 27 | 9 | 28 | 11 | 29 | 8 | 0.9177 | 0.9335 | 0.9253 | 0.8912 | 0.8669 | 0.8931 |
| 70 | 28 | 9 | 29 | 14 | 30 | 13 | 0.9076 | 0.9252 | 0.9250 | 0.8998 | 0.8830 | 0.8841 |
| 71 | 28 | 9 | 29 | 11 | 31 | 9 | 0.9143 | 0.9305 | 0.9251 | 0.8941 | 0.8878 | 0.9008 |
| 72 | 29 | 9 | 29 | 10 | 30 | 9 | 0.9040 | 0.9294 | 0.9271 | 0.9147 | 0.8927 | 0.8979 |
| 73 | 29 | 9 | 30 | 11 | 31 | 10 | 0.9109 | 0.9276 | 0.9262 | 0.9092 | 0.8882 | 0.8943 |
| 74 | 29 | 9 | 30 | 10 | | | 0.9163 | 0.9264 | 0.9264 | 0.9041 | 0.8941 | 0.8941 |
| 75 | 30 | 9 | 31 | 12 | 32 | 9 | 0.9076 | 0.9302 | 0.9254 | 0.9115 | 0.8978 | 0.9076 |
| 76 | 30 | 9 | 31 | 11 | 34 | 8 | 0.9132 | 0.9362 | 0.9252 | 0.9067 | 0.8961 | 0.9155 |
| 77 | 31 | 9 | 31 | 10 | 33 | 9 | 0.9042 | 0.9322 | 0.9274 | 0.9243 | 0.9025 | 0.9142 |
| 78 | 31 | 9 | 32 | 11 | 33 | 8 | 0.9100 | 0.9336 | 0.9255 | 0.9197 | 0.8966 | 0.9182 |
| 79 | 32 | 9 | 33 | 13 | 37 | 11 | 0.9009 | 0.9275 | 0.9251 | 0.9262 | 0.9094 | 0.9157 |
| 80 | 32 | 9 | 33 | 11 | 35 | 9 | 0.9069 | 0.9310 | 0.9255 | 0.9218 | 0.9126 | 0.9238 |
| 81 | 33 | 9 | 33 | 10 | | | 0.8975 | 0.9269 | 0.9269 | 0.9370 | 0.9181 | 0.9181 |
| 82 | 33 | 9 | 34 | 11 | 36 | 10 | 0.9038 | 0.9284 | 0.9254 | 0.9329 | 0.9129 | 0.9203 |
| 83 | 34 | 9 | 33 | 10 | 36 | 7 | 0.8942 | 0.9404 | 0.9263 | 0.9385 | 0.9048 | 0.9279 |
| 84 | 34 | 9 | 35 | 11 | | | 0.9006 | 0.9258 | 0.9258 | 0.9346 | 0.9266 | 0.9266 |
| 85 | 34 | 9 | 35 | 11 | 38 | 8 | 0.9057 | 0.9363 | 0.9253 | 0.9310 | 0.9189 | 0.9352 |
| 86 | 35 | 9 | 35 | 10 | 36 | 9 | 0.8975 | 0.9294 | 0.9273 | 0.9440 | 0.9254 | 0.9295 |
| 87 | 35 | 9 | 35 | 10 | 38 | 8 | 0.9027 | 0.9359 | 0.9255 | 0.9406 | 0.9196 | 0.9369 |
| 88 | 36 | 9 | 37 | 12 | 38 | 10 | 0.8944 | 0.9276 | 0.9253 | 0.9454 | 0.9319 | 0.9362 |
| 89 | 36 | 9 | 37 | 11 | 39 | 9 | 0.8998 | 0.9316 | 0.9261 | 0.9421 | 0.9317 | 0.9411 |
| 90 | 37 | 9 | 38 | 12 | | | 0.8913 | 0.9252 | 0.9252 | 0.9533 | 0.9318 | 0.9318 |
| 91 | 37 | 10 | 37 | 10 | 39 | 9 | 0.9314 | 0.9314 | 0.9268 | 0.9321 | 0.9321 | 0.9413 |
| 92 | 38 | 10 | 39 | 13 | 41 | 12 | 0.9221 | 0.9262 | 0.9254 | 0.9381 | 0.9389 | 0.9413 |
| 93 | 38 | 10 | 39 | 11 | 40 | 10 | 0.9291 | 0.9270 | 0.9255 | 0.9331 | 0.9425 | 0.9449 |
| 94 | 39 | 10 | 39 | 11 | 42 | 8 | 0.9196 | 0.9365 | 0.9256 | 0.9473 | 0.9365 | 0.9500 |
| 95 | 39 | 10 | 39 | 10 | | | 0.9268 | 0.9268 | 0.9268 | 0.9428 | 0.9428 | 0.9428 |
| 96 | 39 | 10 | 39 | 10 | 41 | 8 | 0.9327 | 0.9327 | 0.9254 | 0.9382 | 0.9382 | 0.9502 |
| 97 | 40 | 10 | 41 | 12 | 42 | 9 | 0.9246 | 0.9303 | 0.9254 | 0.9435 | 0.9459 | 0.9520 |
| 98 | 40 | 10 | 41 | 11 | 44 | 9 | 0.9306 | 0.9322 | 0.9251 | 0.9391 | 0.9464 | 0.9556 |
| 99 | 41 | 10 | 42 | 12 | 43 | 10 | 0.9223 | 0.9282 | 0.9259 | 0.9518 | 0.9457 | 0.9516 |

(*Continued*)

**Table 1.** (Continued)

| | Anhøj | | Best box | | Cut box | | Specificity | | | Sensitivity | | |
|---|---|---|---|---|---|---|---|---|---|---|---|---|
| N | C | L | C | L | Cbord | Lbord | Anhøj | Best box | Cut box | Anhøj | Best box | Cut box |
| 100 | 41 | 10 | 41 | 10 | 42 | 9 | 0.9285 | 0.9285 | 0.9265 | 0.9478 | 0.9478 | 0.9510 |

N = number of data points in chart. C = lower limit for number of crossings, L = upper limit for longest run, for declaring random variation by the Anhøj and best box rules. Cbord and Lbord = Additional information for the cut box rules. When specified, parts of the border of the best box to retain to declare random variation. When not specified, cut box is identical to best box (see text for details). Specificity = true negative proportion (no shift). Sensitivity = true positive proportion (shift = 0.8 SD).

### Best box and cut box adjustments to improve the Anhøj rules

To fix some terms, we define a box as a rectangular region $C \geq c$, $L \leq l$ that may be used to define random variation. The corner of the box is its upper right cell $C = c$, $L = l$. In Fig 3 the box $C \geq 2$, $L \leq 6$, marked with red, specifies the Anhøj rules for N = 11. The corner of this box is the cell $C = 2$, $L = 6$.

Based on the crossrun package, which we described in detail in our previous article [9], we developed two functions, bestbox() and cutbox() that automatically seek to adjust the critical values for C and L to balance between sensitivity and specificity requirements. Specifically, the bestbox() function finds the box with highest sensitivity for a pre-determined shift (the target shift), among boxes with specificity $\geq$ a pre-determined value (the target specificity). The cutbox() function subsequently cuts cells from the topmost horizontal and rightmost vertical borders of the best box, starting from the corner while keeping specificity $\geq$ its target value, and the sensitivity for the target shift as large as possible. The result of cutbox() is not necessarily a box, but still a reasonable region for declaring random variation where the corner itself, possibly together with one or more of its neighbours downwards or to the left, may be removed from the best box.

In this study we used a target specificity of 0.925, which is close to the actual average specificity for the Anhøj rules for N = 10–100 and a target shift of 0.8.

Fig 3 illustrates these principles for a run chart with 11 data points. Thus, for N = 11, the Anhøj rules would signal a shift if $C < 2$ or $L > 6$; best box would signal if $C < 3$ or $L > 7$; and cut box would signal if $C < 3$ or $L > 7$, and also when $C = 3$ and $L = 7$.

The following notation is introduced to describe the cut box rules (Table 1): In the rightmost vertical border of the best box $(L = l)$ the part retained within the cut box is stated as $C \geq$ Cbord. Similarly, in the topmost horizontal border of the best box $(C = c)$ the part retained within the cut box is stated as $L \leq$ Lbord. For N = 11, Cbord = 4 and Lbord = 6 (Fig 3 and Table 1), in which case only the corner is cut. If no cut is done, Cbord and Lbord are not specified, these are the cases in which the cut box is identical to the best box.

## Results

We calculated the limits for the Anhøj, best box, and cut box rules together with their corresponding positive test proportions and likelihood ratios for N = 10–100 and SD = 0–3 (in 0.2 SD increments). The limits, specificities, and sensitivities (for SD = 0.8) are presented in Table 1. The R code to reproduce the full results set and the figures from this article is provided in the S1 File_crossrunbox.R. Note that to preserve numerical precision, the code stores the log of likelihood ratios. To get the actual likelihood values back, use exp(log-likelihood).

Fig 2 illustrates the effect of the best box and cut box procedures on the specificity of the runs analysis. As expected, the variability in specificity with varying N is markedly reduced and kept above and closer to the specified target–more with cut box than with best box.

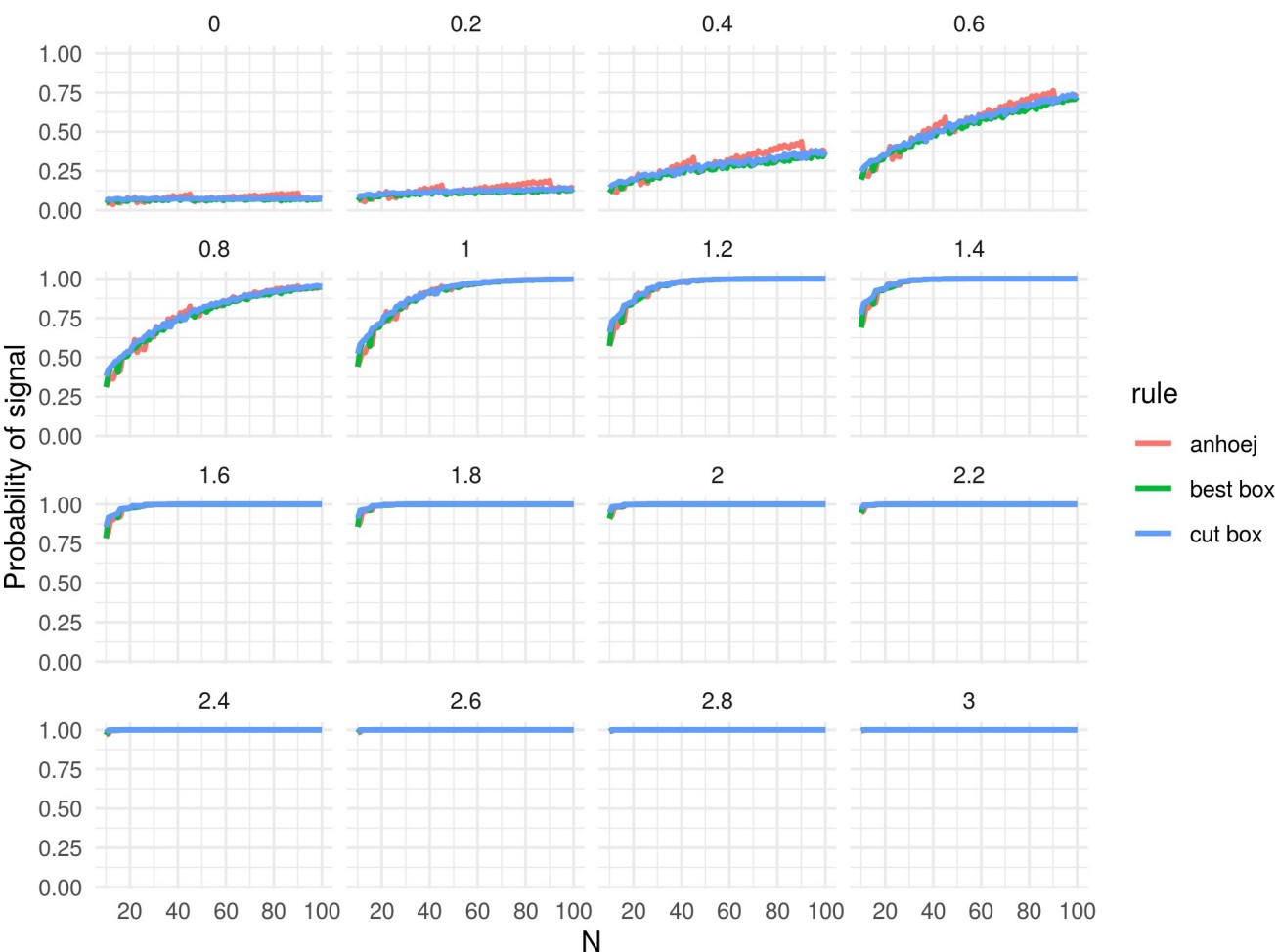

**Fig 4. Power function of Anhøj, best box, and cut box rules.** N = number of data points in run chart. Numbers above each facet represent the size of the shift in standard deviation units (SD) that is present in data.

Fig 4 shows the probabilities of getting a signal as a function of N and SD. The upper left facet (SD = 0) contains the same data as Fig 2. As expected and shown previously in our simulation studies, the power of the runs analysis increases with increasing N and SD [1–3]. The smoothing effect of best box and cut box appears to wear off as N and SD increases. Fig 5 is a blown up version of the facet with shift = 0.8 SD from Fig 4 and shows the sensitivity for the target value used in the box calculations. Exact values for shift = 0 and shift = 0.8 are presented in Table 1

Figs 6 and 7 compare the positive and negative likelihood ratios of the Anhøj rules to the box adjustments. The smoothing effect appear to be of practical value only for positive tests.

## Discussion and conclusion

Based on procedures suggested in our previous paper [9], this study provides exact values for the diagnostic properties of the Anhøj rules for run charts with 10–100 data points including shifts up to 3 standard deviation units.

To our knowledge, and with the exception of our previous work, the properties of the joint distribution of number of crossings and longest runs in random data series have not been studied before.

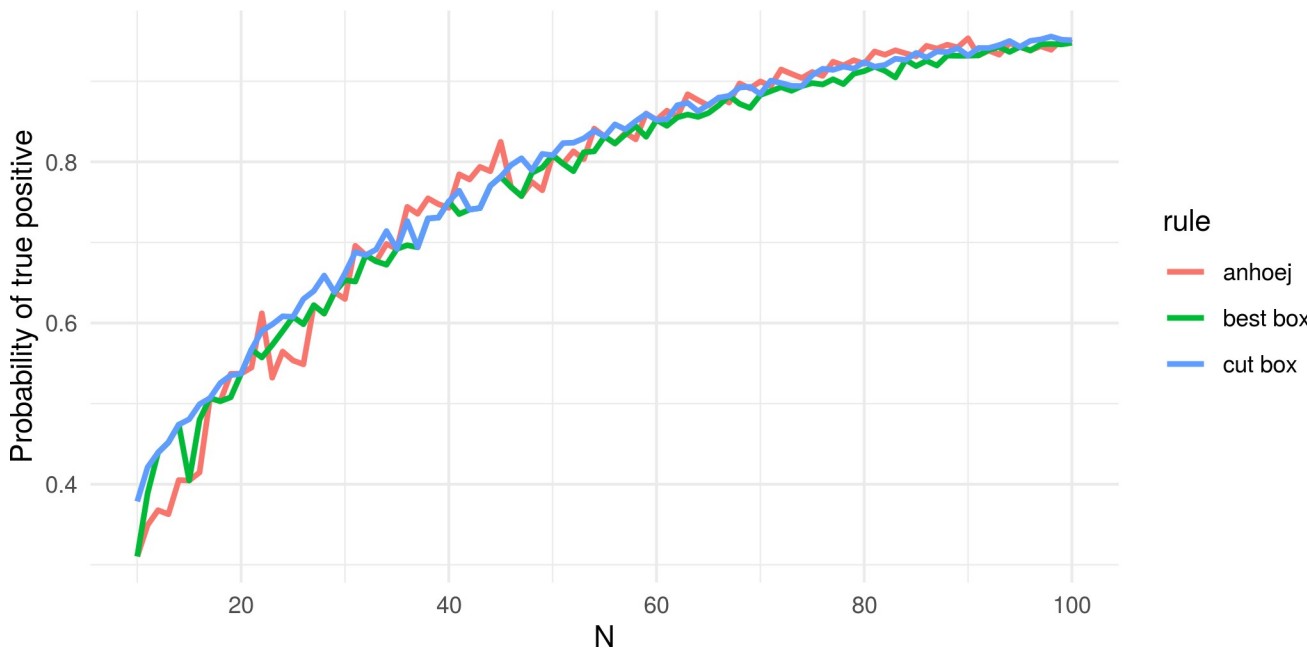

**Fig 5. Sensitivity of Anhøj, best box, and cut box rules for shift = 0.8 standard deviation units.** N = number of data points in run chart.

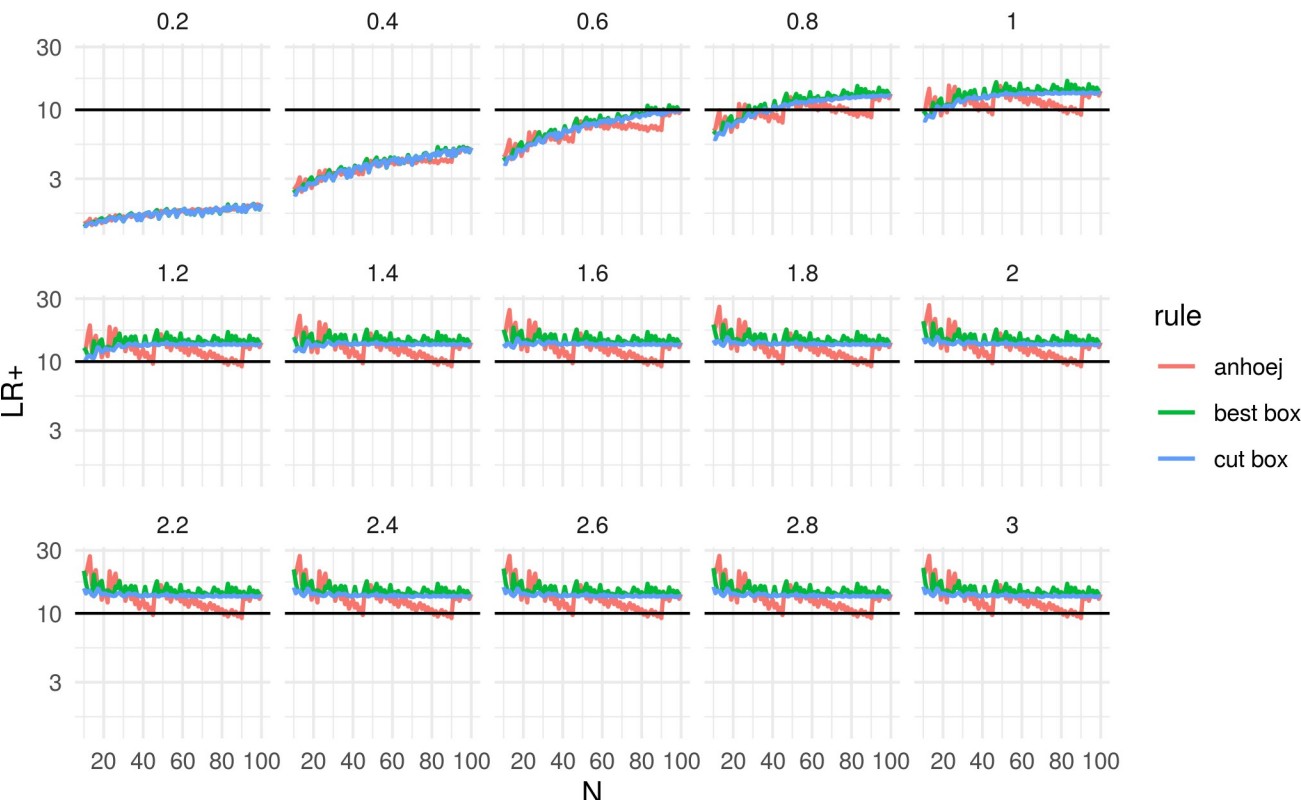

**Fig 6. Positive likelihood ratio of Anhøj, best box, and cut box rules.** N = number of data points in run chart. Numbers above each facet represent the size of the shift in standard deviation units that is present in data.

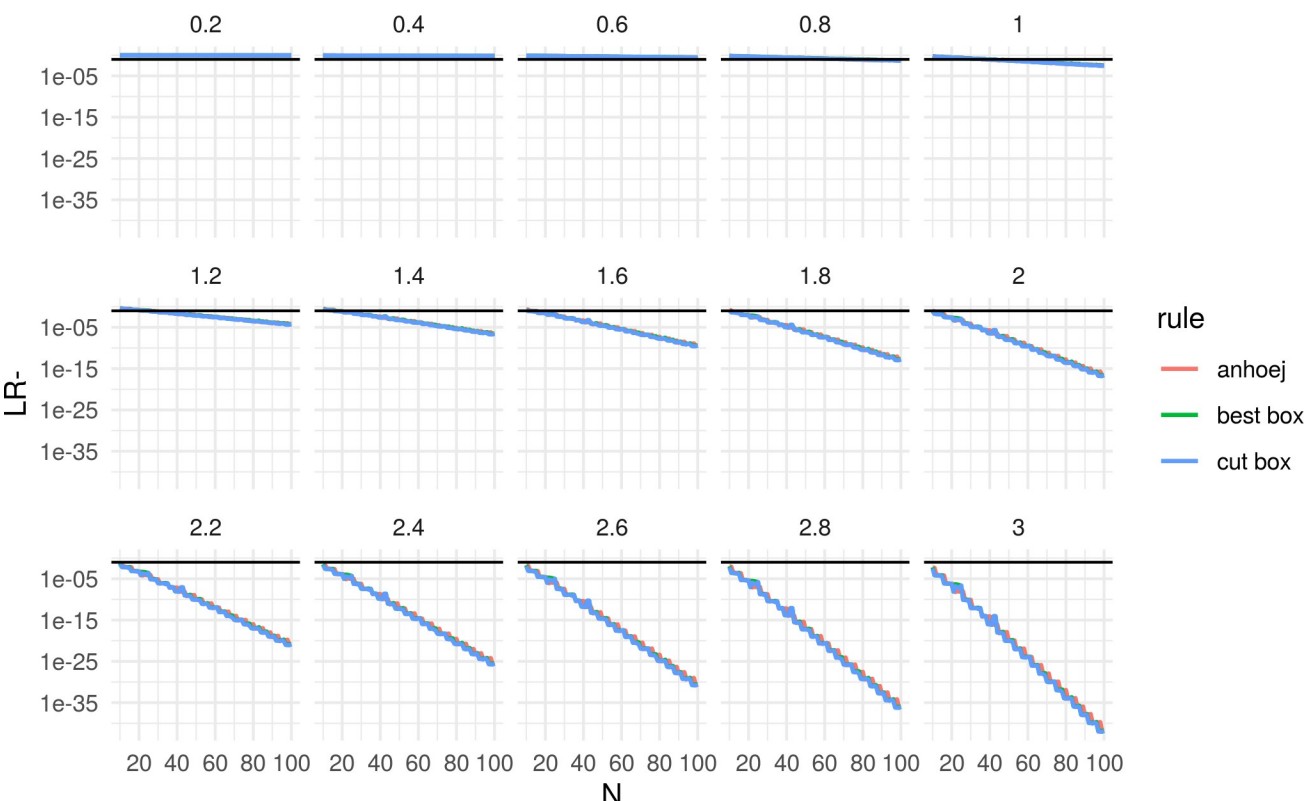

**Fig 7. Negative likelihood ratio of Anhøj, best box, and cut box rules.** N = number of data points in run chart. Numbers above each facet represent the size of the shift in standard deviation units (SD) that is present in data.

Furthermore, the study demonstrates that it is feasible to reduce the variability in run chart specificity with varying number of data points by using the best box and cut box adjustments of the Anhøj rules.

Most importantly, Figs 6 and 7 confirm our experience from years of practical use of runs analysis, that the Anhøj rules constitute a useful and robust method for detection of persistent shifts only slightly larger than 1 standard deviation units and with as little as 10–12 data points. This can be seen by the fact that LR+ > 10 for SD > 1 and N ≥ 10. Although, the best box and cut box procedures will not change this, the box adjustments may improve the practical value of runs analysis by reducing sudden shifts in sensitivity and specificity when the number of available data points changes. Whether this holds true in practice remains to be confirmed.

The study has two important limitations. First, the calculations of box probabilities require that the joint distribution of the number of crossings and longest run is known. As shown in [9] this is the case when the process centre is fixed and known in advance, for example, the median from historical data. In practice the centre line is often determined from the actual data in the run chart, in which case the calculations of box probabilities do not apply. Preliminary studies suggest that this is mostly relevant for short data series. We plan to include a function in a future update of crossrun to calculate the box probabilities with empirical centre lines.

Second, the procedures have so far only been checked for up to 200 data points as detailed in [9]. Because of the iterative procedures and use of high precision numbers using functions from the Rmpfr R package [11] to calculate the joint distributions for varying N, the computations are time consuming, and for N > 100 the precision had to be increased. On a laptop with an Intel Core i5 processor and 8 GB RAM, it takes about one hour to complete

S1_crossrunbox.R for N = 10–100 and SD = 0–3, and the objects created consume over 6 GB of memory. We have no reason to believe that the procedures are not valid for higher N, but the application of the box procedures for larger N may be impractical at the moment.

Also, one should be aware that the value of the box procedures rely on the choice of target specificity and target shift values. Other target values will give different diagnostic properties. Preliminary studies suggest that increasing the target specificity to, say, 0.95 in fact increases the positive likelihood ratios a bit without affecting the negative likelihood ratios considerably. By supplying the R code, we encourage readers to adapt our findings to their own needs.

Regarding the practical application of the box adjustment of the Anhøj rules, we are in the process of testing a method argument for the qic() function from the qicharts2 package that allows the user to choose between "anhoej", "bestbox", and "cutbox" methods to identify non-random variation in run and control charts with up to 100 data points. This will allow us and others to quickly gain practical experience with box adjustments on real life data.

In conclusion, this study provides exact values for the diagnostic properties of the Anhøj rules for run charts with 10–100 data points including shifts up to 3 standard deviation units, and demonstrates that it is feasible to reduce the variability in run chart specificity from varying numbers of data points by using the best box and cut box adjustments of the Anhøj rules.

## Supporting information

**S1 File.**
(R)

## Author Contributions

**Conceptualization:** Jacob Anhøj, Tore Wentzel-Larsen.

**Data curation:** Jacob Anhøj, Tore Wentzel-Larsen.

**Formal analysis:** Jacob Anhøj, Tore Wentzel-Larsen.

**Investigation:** Jacob Anhøj, Tore Wentzel-Larsen.

**Methodology:** Jacob Anhøj, Tore Wentzel-Larsen.

**Project administration:** Jacob Anhøj, Tore Wentzel-Larsen.

**Resources:** Tore Wentzel-Larsen.

**Software:** Jacob Anhøj, Tore Wentzel-Larsen.

**Validation:** Jacob Anhøj, Tore Wentzel-Larsen.

**Visualization:** Jacob Anhøj, Tore Wentzel-Larsen.

**Writing – original draft:** Jacob Anhøj, Tore Wentzel-Larsen.

**Writing – review & editing:** Jacob Anhøj, Tore Wentzel-Larsen.

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
