## [Decision Letter · Decision Letter 0]

20 Mar 2020

PONE-D-20-02344

Smooth operator: Modifying the Anhøj rules to improve runs analysis in statistical process control

PLOS ONE

Dear Dr. Anhøj,

Thank you for submitting your manuscript to PLOS ONE. After careful consideration, we feel that it has merit but does not fully meet PLOS ONE’s publication criteria as it currently stands. Therefore, we invite you to submit a revised version of the manuscript that addresses the points raised during the review process.

Both reviewers think that the problem you are considering is very interesting, although they have very different opinions on the clarity of your explanations. To be more specific, I think that you need more detailed explanations on the statistical methodology. Please revise carefully your exposition in your revised version of the paper.

We would appreciate receiving your revised manuscript by May 04 2020 11:59PM. To enhance the reproducibility of your results, we recommend that if applicable you deposit your laboratory protocols in protocols.io, where a protocol can be assigned its own identifier (DOI) such that it can be cited independently in the future. For instructions see: http://journals.plos.org/plosone/s/submission-guidelines#loc-laboratory-protocols

We look forward to receiving your revised manuscript.

Kind regards,

Miguel Alejandro Fernández, Ph.D.

Academic Editor

PLOS ONE

Journal Requirements:

Reviewers' comments:

Reviewer's Responses to Questions

**Comments to the Author**

1. Is the manuscript technically sound, and do the data support the conclusions?

Reviewer #1: Partly

Reviewer #2: Yes

2. Has the statistical analysis been performed appropriately and rigorously? 

Reviewer #1: No

Reviewer #2: Yes

3. Have the authors made all data underlying the findings in their manuscript fully available?

Reviewer #1: No

Reviewer #2: Yes

4. Is the manuscript presented in an intelligible fashion and written in standard English?

Reviewer #1: No

Reviewer #2: Yes

5. Review Comments to the Author

Reviewer #1: Even though the problem the authors dealt in this paper is very interesting and have practical importance, the methodology of the proposed method is not explained clearly. For example, in method section, authors proposed likelihood ratio, without giving any details. Since this a statistical journal, author must clearly define the random variables C and L and it distribution and how the likelihood ratio arrived. Without this, the methodology proposed is not at all convincing.

Without a well explained methodology, it is hard to rate this paper.

Reviewer #2: this study demonstrates that it is possible to obtain better diagnostic properties of run charts by making minor adjustment to the critical values for C and L. Can this method work well if the distribution of C and L is not known?

6. PLOS authors have the option to publish the peer review history of their article (what does this mean?). If published, this will include your full peer review and any attached files.

Reviewer #1: No

Reviewer #2: No

---

## [Author Response · Author response to Decision Letter 0]

22 Mar 2020

Dear Editor,

Thank you for the opportunity to improve our manuscript. We have revised the article and we believe that we have fully addressed all points raised during the review process.

Here are our responses to the specific issues raised by the reviewers:

Reviewer #1: Even though the problem the authors dealt in this paper is very interesting and have practical importance, the methodology of the proposed method is not explained clearly. For example, in method section, authors proposed likelihood ratio, without giving any details. Since this a statistical journal, author must clearly define the random variables C and L and it distribution and how the likelihood ratio arrived. Without this, the methodology proposed is not at all convincing.

Without a well explained methodology, it is hard to rate this paper.

Response: We have expanded the methods section detailing how to calculate specificity, sensitivity, and likelihood ratios of run chart rules. Also we have made it clear that theses measures have been explained in more detail in previous papers.

Also, reviewer #1 states that all data underlying the findings has NOT been made fully available. This we do not understand. All data including the R code to reproduce our findings are available in the supplementary material S1_crossrunbox.R. 

Reviewer #2: this study demonstrates that it is possible to obtain better diagnostic properties of run charts by making minor adjustment to the critical values for C and L. Can this method work well if the distribution of C and L is not known?

Response: The joint distribution of C and L is, in fact, known in our setting as described in our previous article (Wentzel-Larsen and Anhøj 2019, PLOS ONE). We have added a sentence clarifying this in the Discussion and conclusion section.

---

## [Decision Letter · Decision Letter 1]

15 May 2020

Smooth operator: Modifying the Anhøj rules to improve runs analysis in statistical process control

PONE-D-20-02344R1

Dear Dr. Anhøj,

We are pleased to inform you that your manuscript has been judged scientifically suitable for publication and will be formally accepted for publication once it complies with all outstanding technical requirements.

With kind regards,

Miguel Alejandro Fernández, Ph.D.

Academic Editor

PLOS ONE

Additional Editor Comments (optional):

Reviewers' comments:

Reviewer's Responses to Questions

**Comments to the Author**

1. If the authors have adequately addressed your comments raised in a previous round of review and you feel that this manuscript is now acceptable for publication, you may indicate that here to bypass the “Comments to the Author” section, enter your conflict of interest statement in the “Confidential to Editor” section, and submit your "Accept" recommendation.

Reviewer #1: All comments have been addressed

2. Is the manuscript technically sound, and do the data support the conclusions?

Reviewer #1: Yes

3. Has the statistical analysis been performed appropriately and rigorously? 

Reviewer #1: Yes

4. Have the authors made all data underlying the findings in their manuscript fully available?

Reviewer #1: Yes

5. Is the manuscript presented in an intelligible fashion and written in standard English?

Reviewer #1: Yes

6. Review Comments to the Author

Reviewer #1: (No Response)

7. PLOS authors have the option to publish the peer review history of their article (what does this mean?). If published, this will include your full peer review and any attached files.

Reviewer #1: No

---

## [Editor Report · Acceptance letter]

21 May 2020

PONE-D-20-02344R1 

Smooth operator: Modifying the Anhøj rules to improve runs analysis in statistical process control 

Dear Dr. Anhøj:

I am pleased to inform you that your manuscript has been deemed suitable for publication in PLOS ONE. Congratulations! Your manuscript is now with our production department. 

With kind regards,

on behalf of

Dr Miguel Alejandro Fernández 

Academic Editor

PLOS ONE